# LEARNING TO ENCODE SPATIAL RELATIONS FROM NATURAL LANGUAGE

## ABSTRACT

Natural language processing has made significant inroads into learning the semantics of words through distributional approaches, however representations learnt via these methods fail to capture certain kinds of information implicit in the real world. In particular, spatial relations are encoded in a way that is inconsistent with human spatial reasoning and lacking invariance to viewpoint changes. We present a system capable of capturing the semantics of spatial relations such as behind, left of, etc from natural language. Our key contributions are a novel multi-modal objective based on generating images of scenes from their textual descriptions, and a new dataset on which to train it. We demonstrate that internal representations are robust to meaning preserving transformations of descriptions (paraphrase invariance), while viewpoint invariance is an emergent property of the system.

## 1  INTRODUCTION

Through natural language humans are able to evoke representations in each others' minds. When one person describes their view of a scene their interlocutors are able to form a mental model of the situation and imagine how the objects described would look from different viewpoints. At the simplest level, if someone in front of you describes an object as situated to their left, you understand that it is to your right. Current models for embedding the meaning of natural language are not able to achieve such viewpoint integration. In fact, as shown by Gershman & Tenenbaum (2015), distributed representations of natural language extracted from monolingual corpora fail to understand semantic equivalences such as that 'A is in front of B' describes the same situation as 'B is behind A.'.

In this paper we seek to understand how to build representations that can capture these invariances, an important first step toward a human-level ability to perform abstract spatial reasoning. We hypothesize that to capture spatial invariance in language, representation learning must be grounded in vision. To this end, we introduce a dataset and task designed for the specific purpose of learning and evaluating representations of spatial language. We couple 3D scenes with language descriptions of several different viewpoints of a scene, facilitating the learning of spatial relations, spatial paraphrases and rotational equivalences.

Unlike visual question answering tasks such as CLEVR (Johnson et al., 2017), this task requires systems to read descriptions of a scene, each paired with a camera position, and create an internal model of the described world. They must then demonstrate the efficacy of their model by generating visual representation of the world conditioned on a viewpoint. Training a multi-modal generative model on this dataset, we can learn scene representations in an unsupervised manner. Our approach is to train the model to translate (multiple) natural language descriptions into images, taking into account the viewpoints both of the input (text) and the output (image). Solving this task requires understanding the language of spatial relations.

Finding that the model performs well on the image generation task, we analyse the constituent parts of the model, solve the challenge posed by Gershman & Tenenbaum (2015), and provide a roadmap to incorporating spatial understanding in language models more generally.

The main contributions of this paper are:

- We introduce two new datasets which contain language descriptions (synthetic and natural) and visual renderings of 3D scenes from multiple viewpoints.

- We use those datasets to learn language representations of a scene grounded in the visual domain. We show that the model is able to form a representation of the 3D scene from language input.

- We analyse these representations and demonstrate that they encode spatial relations and their corresponding invariances in a way consistent with human understanding, and that these language-based invariances arise at the level of the language encoder, with additional viewpoint-invariance as an emergent property of the overall system.

## 2 RELATED WORK

We consider two lines of related work: previous efforts towards understanding spatial relations across NLP, Computer Vision and Robotics, and more broadly work in visual question answering where explicit visual reasoning is required.

Spatial natural language is notoriously ambiguous and difficult to process computationally. Even seemingly simple prepositions like *behind* are impossible to describe categorically and require a graded treatment (e.g. how far may a person move from behind a tree before we no longer describe them as such). Likewise, understanding the difference between allocentric and egocentric reference frames can easily confound (Vogel & Jurafsky, 2010; Kranjec et al., 2014). The lexicalisation of spatial concepts can vary widely across languages and cultures (Haun et al., 2011), with added complexity in how humans represent geometric properties when describing spatial experiences (Landau & Jackendoff, 1993) and in the layering of locatives (Kracht, 2002). While there has been considerable research into the relationship between categorical spatial relation processing, perception and language understanding in humans, there are few definite conclusions on how to encode this relationship computationally (Kosslyn, 1987; Johnson, 1990; Kosslyn et al., 1998; Haun et al., 2011).

In the field of Natural Language Processing, work on spatial relations has focused on the extraction of spatial descriptions from text and their mapping into formal symbolic languages (Kordjamshidi et al., 2012a;b), with numerous such annotation schemes and methods proposed (Shen et al., 2009; Bateman et al., 2010; Rouhizadeh et al., 2011). In other related work, Vogel & Jurafsky (2010) grounded spatial language in an environment through reinforcement learning. The action- and state-spaces of the environment however were simplified to the extent that spatial relations were represented by tuples such as west-of$(A, B)$, turning this again into a semantic parsing problem more than anything else.

Meanwhile research on visualising spatial descriptions has predominantly employed heavily hand engineered representations that do not offer the generic cross task advantages of distributed representations (Chang et al., 2014; Hassani & Lee, 2016). In robotics, spatial relations feature in research on human-robot interaction such as (Tellex et al., 2011). Here, similar to Vogel & Jurafsky (2010), spatial description clauses are translated into grounding graphs, with objects represented by bounding boxes and relations by binary, geometric features such as *supports(x,y)*.

There have been a number of datasets that bear some similarity to the dataset produced as part of this work. The SHAPES (Andreas et al., 2016) and CLEVR datasets (Johnson et al., 2017) contain images and (synthetic) language questions that test visual reasoning. CLEVR in particular uses a form of 3D rendering that produces images similar to the ones used here. While on the surface, these tasks may look similar, it is important to note that this paper is not about visual question answering, but rather about the understanding and modelling of spatial relations. This is achieved by making available multiple viewpoints of the same scene and presenting scenes in viewpoint-specific formats. The key results in this work concern the text representations learnt alongside the visual ones rather than the mapping from one to the other.

## 3 DATASET OF VISUALLY GROUNDED SCENE DESCRIPTIONS

We construct virtual scenes with multiple views each presented in multiple modalities: image, and synthetic or natural language descriptions (see Figure 1). Each scene consists of two or three objects placed on a square walled room, and for each of the 10 camera viewpoints we render a 3D view of the scene as seen from that viewpoint as well as a synthetically generated description of the scene. For natural language we selected a random subset of these scenes and had humans describe a given

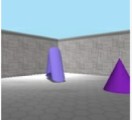
**NL** There is a small purple cone far to the right slightly cut off at the edge of the screen. There is a lighter purple object that almost looks like a sideways outlined triangle slightly farther away near the center of the screen.
**SYN** There is a violet cone to the far right of a purple torus. The cone is in front of the torus.

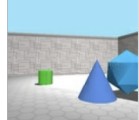
**NL** A blue cone is sitting near the right side of the screen. To the right of it and slightly behind is a 3d polygon sharp edged object, slightly cut off by the right edge of the screen. In the background and to the left of these objects is a green cube.
**SYN** A blue icosahedron is in front of a right green cylinder. There is a green cylinder far left of a blue cone. The icosahedron is close to the cone.

**Figure 1:** Example descriptions with corresponding ground truth image. Shown are natural (NL) and synthetic (SYN) language descriptions. Annotation linguistic errors have been preserved.

scene and viewpoint based on the rendered image. Details on data generation are found in Appendix B. We targeted a fairly low visual complexity of the rendered scenes since this factor is orthogonal to the spatial relationships we want to learn. The generated scenes still allow for large linguistic variety (see Table 1). We publish the datasets described in this paper at `https://anonymous`.

**Synthetic language data:** We generated a dataset of 10 million 3D scenes. Each scene contains two or three coloured 3D objects and light grey walls and floor. The language descriptions are generated programmatically, taking into account the underlying scene graph and camera coordinates so as to describe the spatial arrangement of the objects as seen from each viewpoint.

**Table 1:** Dataset statistics.

|  | Synthetic | Natural |
|---|---|---|
| # Training Scenes | 10M | 5,604 |
| # Validation Scenes | 1M | 432 |
| # Test Scenes | 1M | 568 |
| Vocabulary Size | 42 | 1,023 |
| Tokens per Description | 60 | 90 |

**Natural language data:** We generated further scenes and used Amazon Mechanical Turk to collect natural language descriptions. We asked annotators to describe the room in an image as if they were describing the image to a friend who needs to draw the image without seeing it. We asked for a short or a few sentence description that describes object shapes, colours, relative positions, and relative sizes. We provided the list of object names together with only two examples of descriptions, to encourage diversity, while focusing the descriptions on the spatial relations of the objects. The annotators annotated 6,604 scenes with 10 descriptions each, one for each view.

## 4 SCENE ENCODING TASK & MODEL

We use this dataset in a scene encoding experiment, where the task is to draw an image given a number of written descriptions of a scene. Considering our overarching goal of learning to encode spatial language from text, this task has multiple desirable properties. Having multiple descriptions per scene encourages learning paraphrases, and particularly those of spatial language. By combining input descriptions with viewpoints and requiring the model to generate outputs from different perspectives, this is reinforced, and in addition learning of viewpoint invariant representations is encouraged. This is further encouraged by setting up the task in such a way that the encoder does not have access to the output viewpoint, which puts pressure on the model to encode the descriptions into a viewpoint invariant representation.

For the experiments presented in this paper, we adapted the model by Eslami et al. (2018), the main difference to their model being the use of language encoders, rather than visual ones. While a number of models could plausibly be used for the work presented here, their setup particularly lent itself to integrating information from multiple viewpoints. This is useful as we want to learn a viewpoint invariant representation for a single scene. We provide full details of the model implementations in Appendix A, and refer to the model as the Spatial Language Interpretation Model (SLIM), when in need of an acronym. On a high level, the model consists of two parts. First an encoder combines all input descriptions and their viewpoints to learn a single vector representation of a scene. Second, this representation is fed into a conditional generative model (DRAW, Gregor et al. (2015)) to produce an output image. See Figure 2 for a schematic description of the model.

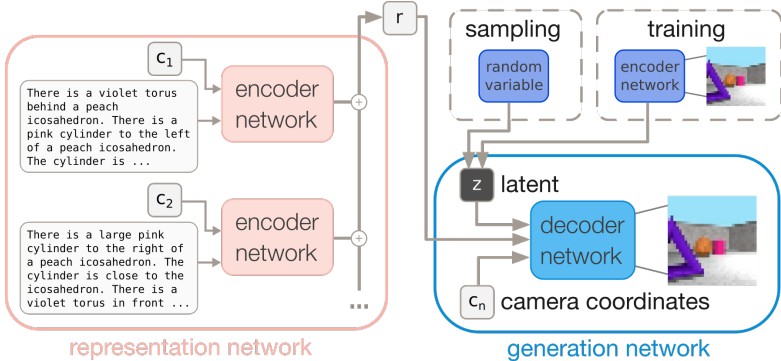

**Figure 2:** Diagram of model used for experiments. A representation network parses multiple descriptions of a scene from different viewpoints by taking in camera coordinates and a textual caption. The representations for each viewpoint are aggregated into a scene representation vector $r$ which is then used by a generation network to reconstruct an image of the scene as seen from a new camera coordinate.

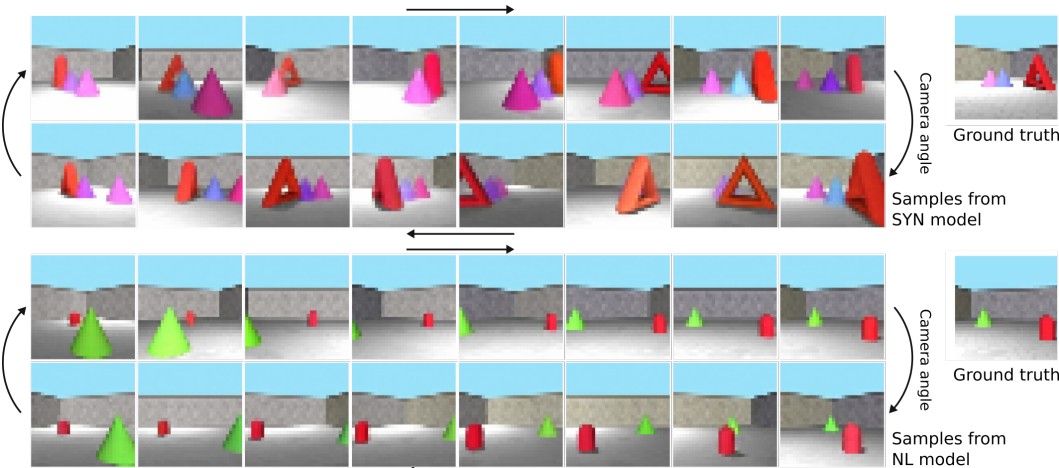

**Figure 3:** Samples generated from the synthetic (top) and natural language (bottom) model. Corresponding captions are: "There is a pink cone to the left of a red torus. There is a pink cone close to a purple cone. The cone is to the left of the cone. There is a red torus to the right of a purple cone."; "There are two objects in the image. In the back left corner is a light green cone, about half the height of the wall. On the right side of the image is a bright red capsule. It is about the same height as the cone, but it is more forward in the plane of the image."

### 4.1    A NOTE ON EVALUATION

While pixel loss compared with gold images is an obvious metric to optimise against when generating images, human judgements are more suitable for evaluating this task. More precisely, pixel loss is too strict a metric for evaluating the generative model due to the differing degrees of specificity between language and visual data points. Consider the images and descriptions in Figure 1. Adjectives such as *big* or relative positions such as *to the far right* are sufficient to give us a high-level idea of a scene, but lack the precision required to generate an exact copy of a given image. The same description can be satisfied by an infinite number of visual renderings—implying that pixel loss with a gold image is not a precise measure of whether a given image is semantically consistent with the matching scene description. Human judgements on the other hand allow us to evaluate this task on the desired property, preferring spatial consistency over pixel-accuracy in the generated images.

We showed annotators model samples with corresponding language descriptions, and asked them to judge whether the descriptions matched, partially matched, or did not match the image. We additionally asked for binary choices on whether (1) all object shapes and (2) all object colours from the text are in the image; whether (3) all shape and colour combinations are correct; and

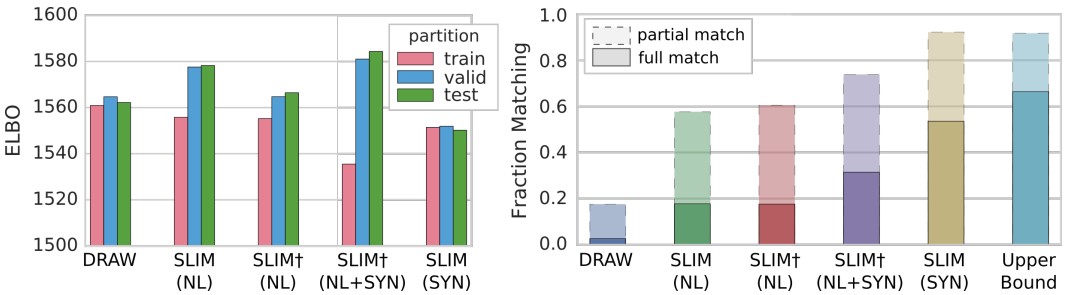

**Figure 4:** On the left, ELBO numbers for the model variants under training for train/validation/test splits. On the right, human ranking of consistency of visual scene samples with matching synthetic caption. For SLIM[†] (NL+SYN) numbers shown were calculated only with natural language inputs.

whether (4) all objects are correctly positioned. Annotators saw randomly mixed samples from all the models and benchmarks. For the UPPER BOUND evaluation we showed gold images and matching descriptions.

Irrespective of the training setup below, we use synthetic descriptions of the output in the human annotation, as these guarantee us a consistent degree of specificity versus natural language descriptions that can widely vary in that respect. Note that this only concerns the description shown to the human annotators. In the evaluation, we feed synthetic language into the model for the experiments in Section 4.2, and natural language descriptions in the case of Section 4.3.

## 4.2 SYNTHETIC LANGUAGE EXPERIMENTS

We trained our model on the synthetic text to verify whether it can be trained to understand scenes from descriptions. For each scene we randomly chose nine out of ten available views as inputs (text), and the final viewpoint as the target (image). Training was done with a batch size of 32 examples, using early stopping based on the validation data.

Figure 3 shows model samples and the underlying ground truth. We draw samples from the model by sampling the latents from the prior and varying the camera position. Note that while the image samples exhibit a lot of variability and can differ widely from the ground truth image, they are consistent with the scene text description.

We report ELBO loss on training, validation and test data in the left part of Figure 4. As a baseline, we train an unconditional DRAW generative model (i.e. no scene descriptions are provided) which learns the unconditional output image distribution. The ELBO captures how well the model can model the distribution of the images of the scene, and the results suggest that all setups under consideration are capable of doing this. However the ELBO does not capture the conditional likelihood given the representation and therefore does not show a significant difference between our conditional setup and the unconditional model.

Figure 4 right shows the results of the human evaluation. Naturally, samples from the unconditional baseline perform poorly as it does not condition on the descriptions used as basis for the human judgements. However, these random samples still serve a purpose in informing us about the complexity of the dataset as well providing a lower bound on the generosity of the human annotators, so to speak.

The UPPER BOUND results are from annotators comparing ground truth images with their matching synthetic description. Note here that with 66.39% perfect matches and 91.70% including partial matches, these results are far from perfect. This indicates several issues: annotators may have used a very strict definition of a perfect match, penalising the fact that the synthetic descriptions are succinct and leaving out details such as the background colours for instance. Gold standard synthetic captions can also be perceived as incorrect due to factors such as the description of relative

locations[1], occlusion, or colour names not matching an annotators understanding (see Appendix B.2 for examples). Our model, SLIM (SYN), matches the performance of the UPPER BOUND in the human judgements, scoring 92.19% including partial matches, while underperforming slightly with respect to perfect matches.

### 4.3 NATURAL LANGUAGE EXPERIMENTS

Following the synthetic language experiments, we investigate whether the model can cope with natural language. Due to cost of annotation, the amount of natural language data is several orders of magnitude smaller than the synthetic data used in the previous section (see Table 1). We consider multiple training regimes that aim to mitigate the risk of over-fitting to this small amount of data. The naïve setup (SLIM (NL) in Figure 4) uses only the natural language training data. We also took a frozen pre-trained generation network (trained on synthetic data) and trained the representation network alone on the same dataset (SLIM$^\dagger$ (NL)). Finally, SLIM$^\dagger$ (NL+SYN) uses the same pre-training and uses an augmented dataset with a 50/50 split between synthetic and natural language. We evaluate all three models with natural language inputs only, independent of training protocol.

We follow the training regime described above for the synthetic data. In addition, we found that adding 50% dropout together with early stopping on validation data provided the best performance. The drastic increase in complexity compared with the synthetic task is evident in the lower scores. However, and more importantly, the relative performance of the three natural language training setups is consistent across all training runs, with the joint model (frozen decoder, encoder trained jointly on synthetic and natural language) performing best and the model trained on natural language only performing the worst (Figure 4). This is encouraging, both as the absolute numbers suggest that the SLIM architecture is capable of encoding spatial relations from natural language, and as the significant gains in the joint training regime over the naïve approach highlight the potential of using this type of data for further research into simulation to real world model adaptation.

## 5 REPRESENTATION ANALYSIS

Having established that we can successfully create a representation of the scene to reconstruct an image view, we now turn our attention to how this representation is built up from language descriptions of the scene. We show that our model does learn semantically consistent representations, unlike the neural network models investigated in Gershman & Tenenbaum (2015). Secondly, we examine representations at different stages of our model to establish at what point this semantic coherence arises, and similarly, at what point viewpoint invariance (independence of input camera angles) is achieved, allowing us to recreate scenes from new viewpoints. The following set of equivalences exemplifies the difference between the two concepts:

| | | | |
|---|---|---|---|
| **Paraphrase invariance** | (A left of B) | $\equiv$ | (B right of A) |
| **Viewpoint invariance** | (A left of B, $0°$) | $\equiv$ | (A right of B, $180°$) |

### 5.1 SEMANTIC COHERENCE ANALYSIS

Gershman & Tenenbaum (2015) consider sentence transformations (Figure 5) and compare human similarity judgements of these transformations, to the base sentence, with the distances of the output of neural network models. That work showed that recurrent language encoding models fail to coherently capture these semantic judgements in output representations.

Human annotators *rank* the transformations of the base sentence, B, in order of semantic similarity, resulting in average rankings $M > P > A > N$, from most to least similar. In Gershman & Tenenbaum (2015) a set of neural language encoders is then shown not to follow this human judgement. In particular, the M transformation—semantically equivalent—is never the most similar to the base sentence. We believe that the original experiment's conclusions may have been too broad: first, as the authors themselves state, the pre-trained language models considered were built on English

---

[1]Consider two objects behind each other with one shifted slightly to the left of the other. Their relationship could be described as "behind and left", just "behind" and under some circumstances also as just "to the left of", with different annotators preferring different schemes and considering others invalid.

Wikipedia and the Reuters RCV1 corpus (Collobert et al., 2011), neither of which exhibit significant amounts of language in the style of the test data. Next, there was nothing in the objective functions for these models that would encourage solving the task at hand. This is the key question: is the deflationary criticism with respect to the model architectures valid, or is the failure to solve this task a function of the training regime?

We posit that a learning objective grounded in another modality—such as the visual domain—could learn spatial semantic coherence by directly teaching the model a complex function to reconcile syntax (the words) and semantics (the meaning, here captured by the scene and its visual representation). We test this hypothesis by re-implementing the sentence transformation setup in our multi-modal setup.

We have four scene templates with synthetic descriptions transformed as in Gershman & Tenenbaum (2015) (Table 1). The meaning preserving transformation, M, naturally results in the same scene as the base scene; however the resultant descriptions match the original transformation from B to M.

For this analysis we use a model trained on synthetic data (see Section 4.2). We separately encode the set of descriptions for base scene ($r$) and transformations ($r_n, r_a, r_p, r_m$) and compute their similarity using the cosine distance between the representations.[2]

| B | Base sentence | "a young woman in front of an old man." |
| N | Noun change | "a young man in front of an old woman." |
| A | Adjective change | "an old woman in front of a young man." |
| P | Preposition change | "a young woman behind an old man." |
| M | Meaning preservation | "an old man behind a young woman." |

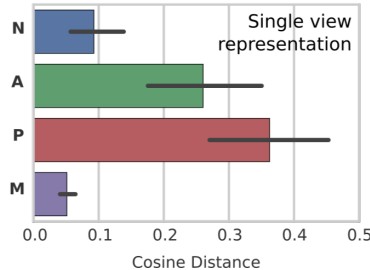

**Figure 5:** Top, transformations following Gershman & Tenenbaum (2015). Bottom, averaged cosine similarity between the representation obtained by encoding a single caption with a fixed camera angle and the transformed representation. The meaning preserving representation is the most similar; followed by noun, adjective and preposition changes. Black bars represent 95% CI, average over 32 scenes.

ity using the cosine distance between the representations.[2] The results are depicted in Figure 5, averaged over 3,200 scenes and their transformations. Unlike the results in Gershman & Tenenbaum (2015), the SLIM architecture encodes B and M as the most similar, both when considering a single sentence encoding and even when considering the full scene representations.

Of the semantically different descriptions, the noun transformations produce the least dissimilar representations. We believe this is due to the fact that the reconstruction loss is based on a pixel measure, where a colour change (adjective change) leads to a larger loss in terms of the overall number of pixels changed in the visual representation than the noun change which only alters the shape of the objects, causing a smaller number of pixels to change in the overall visual representation.

This reinforces our earlier point that in order to properly encode the kind of syntactic/semantic divergence tested for by this analysis, we need models that train not only on syntax but also on semantics. While the visual domain serves as a useful proxy for semantics in our setup, the pixel loss analysis highlights that it is only a proxy with its own shortcomings and biases.

## 5.2 ENCODER HIERARCHY REPRESENTATION ANALYSIS

Here we show that while individual encodings are strongly view-dependent, the aggregation step and decoder integrate these representations into a viewpoint invariant end-to-end model. We investigate how representations for the same scene differ at a number of stages: after encoding the language and camera angle, after the aggregation step, and finally after decoding.

Figure 7a is a t-SNE plot of the intermediate representations for a number of scenes. Each point in that plot corresponds to a single language and coordinate embedding, with embeddings for all 10 views in 32 scenes shown. The embeddings cluster by camera angle even though they are sampled from different scenes. This shows how on a superficial level the camera coordinates dominate the single input representations. Figure 7b reinforces this analysis: If a camera independent representa-

---

[2]Euclidean distance and Pearson's r result in the same ordering of similarities.

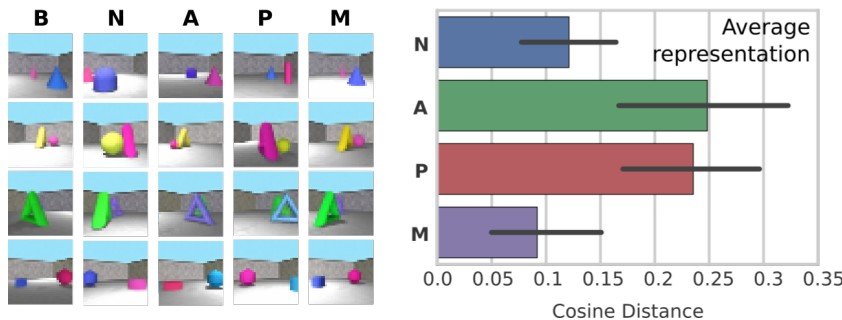

**Figure 6:** Left, samples from the model when fed input contexts transformed according to Gershman & Tenenbaum (2015) transformations. Right, average cosine distance between base representation and aggregated representation induced by applying one of the four transformations to each of the context inputs. Black bars represent 95% CI, average over 32 scenes.

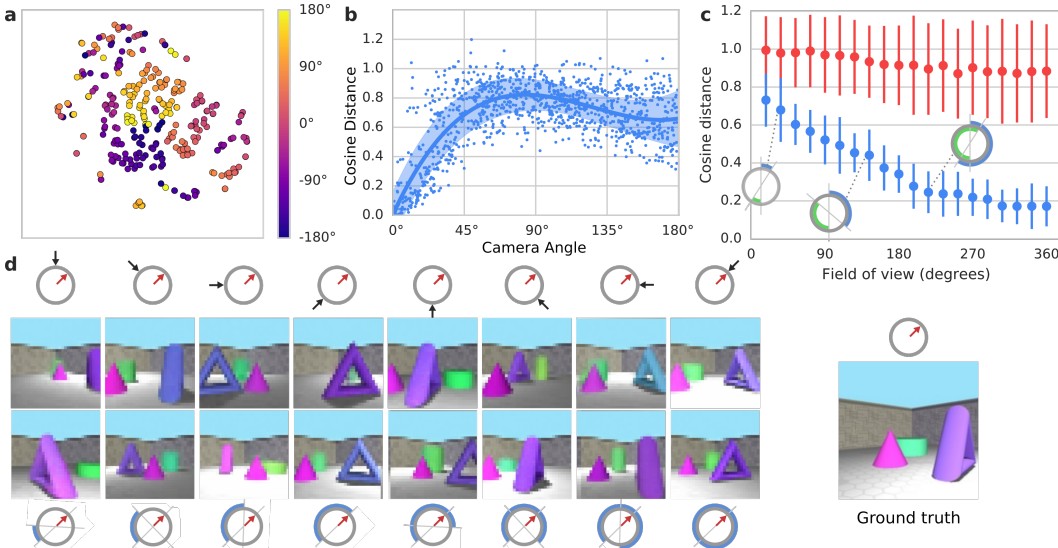

**Figure 7:** a) t-SNE of single description encodings, coloured by camera angle. b) Distance between single description representations of the same scene as a function of the angle between their viewpoints. c) Distance between aggregated representations drawn from opposing arcs as a function of the size of those arcs. Blue compares same scene representations, red different scene representations. d) Output samples for a constant scene and coordinate under varying input conditions. Top: a single description (from black arrow), bottom: aggregated descriptions from an increasingly sized arc.

tion were built at the language encoder level, we would expect each intermediate representations to be similar, with the aggregation process only reducing noise. The quickly diverging cosine distances as a function of the angle between two camera coordinates shows that this is not the case, and it is clear that the representations do not have the desired property at this stage in the model.

Next we investigate the effect of the aggregation function. Consider a circle around a scene, with arcs originating at $0°$ and $180°$, respectively. We clockwise increase the central angle $\theta$ of both arcs simultaneously and calculate aggregated representations of nine language descriptions from within each arc.[3] Figure 7c (blue) shows how the cosine distance between these representations decreases as the size of the two arcs increases. While $\theta \leq \pi$ the two representations are non-overlapping, demonstrating how even the simple additive aggregation function is able to cancel out multiple viewpoints while integrating their information for added robustness. If the representations corresponding to each arc are sampled from two different scenes (red), the representations no longer converge as more data is integrated. Together these analyses suggest that the aggregation step is

---

[3] Sampled with replacement.

doing something beyond cancelling out noise such as cancelling out the non-invariant parts of the encoding.

Figure 7d shows that the model achieves viewpoint invariance. We sample images from the model keeping the output coordinates and scene constant, but varying the information we input. Regardless of the number of input descriptions or their location, we receive images that are broadly consistent with the semantics of the gold scene. As the input view approaches the target or as the number of points sampled increases, the samples become semantically more consistent with the ground truth.

## 6 CONCLUSION

We have presented a novel architecture that allows us to represent scenes from language descriptions. Our model captures both paraphrase and viewpoint invariance, as shown through a number of experiments and manual analyses. Moreover, we demonstrated that the model can integrate these language representations of scenes to reconstruct a reasonable image of the scene, as judged by human annotators. Lastly, we demonstrated how to effectively use synthetic data to improve the performance of our model on natural language descriptions using a domain adaptation training regime.

This paper serves to highlight one key point, namely the importance of aligning training paradigms with the information one wishes to encode. We demonstrated how such information can be provided by merging multiple modalities. More broadly, this also demonstrates that the criticism in Gershman & Tenenbaum (2015) is not an in-principle problem of the model, but rather a matter of setting up your training objectives to match your desired outcomes.

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

# A  MODEL DETAILS

The generative model used for the experiments, depicted in Figure 2, consists of two parts: a representation network that produces an aggregated representation from textual descriptions of the scene from a number of viewpoints, and a generation network conditioned on the scene representation that renders the scene as an image from a new viewpoint. We describe both networks below.

This model follows the framework introduced by Generative Query Networks (Eslami et al., 2018), which has introduced the idea of querying the model to reconstruct a view of the scene from a novel angle to force viewpoint invariance in the representation. This can be used to generate new views of the same environment.

## A.1  REPRESENTATION NETWORK

The representation network is composed of an encoder and an aggregation step. The encoder transforms each text observation $d_i$ and corresponding camera angle $\theta_i$ into a viewpoint embedding $h_i$, $(i = 1 \ldots n)$; and the aggregation combines those embeddings into a single representation vector.

The input to the representation network is a sequence of pairs $(d_i, \theta_i)$. The encoder network consists of a convolutional language encoder over the sequence of embedded words (embedding dimension 64) $\hat{d}_i = \text{CNN}(\text{embed}(d_i))$. In particular, we use a variant of ByteNet (Kalchbrenner et al., 2016) with three 1D convolutional layers, dilation factor of 2 and layer normalisation followed by average aggregation over the sequence dimension. The camera is represented by the embedding $\hat{c}_i = \text{MLP}([\cos(\theta_i), \sin(\theta_i)])$ (dimension 32). Next, these two representations are merged via a three-layer residual MLP to generate an embedding vector $h_i = \text{MLP}(\hat{d}_i \| \hat{c}_i)$ with dimensionality 256.

The aggregation step consists of computing the scene representation $r = \frac{1}{n} \sum_i^n h_i$, independent of the number of inputs. For training $n = 9$. Clearly, more complex functions could be used as the aggregator, but for the purposes of the investigation presented here this relatively simple aggregation is sufficient.

## A.2  GENERATION NETWORK

The generation network is a conditional generative model that learns the distribution of likely images given a representation.

To train the generative model, we sample a target pair $(d_t, \theta_t)$ that was not provided to the representation network. This pair is used to train a conditional autoencoder (Sohn et al., 2015) where the conditioning variable is the concatenation of $r$ and $\theta_t$.

We use the DRAW (Gregor et al., 2015) network to implement the generation network. DRAW is a recurrent variational autoencoder which has been used successfully to render images of complex scenes such as the 3D images in the dataset. We use a DRAW model with 12 iterations and with a convolutional LSTM core with dimensionality 128. The conditioning variable is concatenated with the sampled latent $z$ at each iteration. The output distribution is a Bernoulli on each subpixel. We train the model by minimizing the ELBO (Kingma & Welling, 2013), which is made up of the total likelihood of the target image under this distribution plus the KL for the DRAW prior:

$$\mathcal{L} = -\log D(x|r) + \sum_{k=1}^{K} KL\left(Q(z_k|h_k^{enc}) \| P(z_k)\right)$$

where $x$ is the image to be reconstructed and $r$ is the representation created by the encoders. The KL is the sum of terms from each of the $K$ iterations in draw with $z_k$ denoting each latent for an iteration and $h_k^{enc}$ representing the encoded latent.

The reason for using the autoencoding component of the model is to guide the training, while simultaneously constraining it through the KL-term on an uninformative prior (in our case a zero mean unit variance gaussian). While the autoencoder can help kick-start training, the model is strongly encouraged to decrease its reliance on $z$ and instead to extract all necessary information from $r$.

## A.3 TRAINING DETAILS

We train our model using the ADAM optimizer with a learning rate annealing schedule starting at $5e - 4$ and decaying linearly to $5e - 5$ over one million steps. Training is stopped at the minimum validation loss calculated every 500 steps for 3200 samples (100 minibatches). For the synthetic dataset we use dropout of 0% while for natural language we use 50% dropout.

## B DATASET GENERATION DETAILS

To generate the synthetic scenes we employed the MuJoCo engine[4] to create a generic 3D square room where multiple simple geometric objects are placed at random. The room is represented as a square with coordinates $x, y \in [-1, 1]$, with object coordinates sampled uniformly. Each scene contains two or three objects the identities of which are determined by a set of three latent variables: 8 shapes[5]; a HSV color sampled uniformly in the intervals $H \in [0, 1]$; $S \in [0.5, 1]$; $V \in [0.8, 1]$; and object size [6]. Lastly, we randomly sample camera positions from a circle centred at the middle of the room, with a radius approximately equal to the distance from the walls to the center.

To create a data batch we sample 10 camera positions and for each we render a view of the scene from that angle and associate with it a textual description. The generated images are $128 \times 128$ RGB images which are downsampled to $32 \times 32$ and rescaled to floating point values in the range $[0, 1]$ before being fed to the model.

The first part of our dataset contains synthetic language descriptions where the descriptions are generated by a script with access to the scene geometry. We iterate over object pairs, and for each object determine its shape name (matching the shapes described above, a color name (found by looking up the nearest HSV neighbor in a look-up table with 22 named colours) and a size name (large, small or no size). A description is then sampled by generating a caption relating those two objects with a spatial relation (in front of, behind, left of, right of, close, far). The order of objects in the description is randomly sampled. The final caption is composed of enough such descriptions so as to cover every pairwise relation for all objects in the scene exactly once.

To generate the natural language dataset we subsampled scenes from the synthetic dataset and asked humans to write a description of the rendered image of the scene from a certain camera position.

We have created validation and test sets based on held out combinations of colour and object type for each of the synthetic and natural language labeled scenes, with different combinations held out in the validation and the test data, and a guarantee that each validation/test scene contains at least one object unobserved during training.

The training set does not contain objects from the set {'yellow sphere', 'aqua icosahedron', 'mint torus', 'green box', 'pink cylinder', 'blue capsule', 'peach cone'}. The validation set scenes contain at least one of the first three elements of the held out combinations set and none of the remaining four elements. The test set scenes can contain any combination of shapes or colours, making sure they contain at least one of the last four combinations in the held out set.

---

[4]http://www.mujoco.org/

[5]cube, box, cone, triangle, cylinder, capsule, icosahedron, sphere

[6]a scaling factor for the object mesh chosen uniformly from an interval sensible for the MuJoCo renderer

## B.1 DATASET EXAMPLES

### B.1.1 SYNTHETIC LANGUAGE, TWO OBJECTS

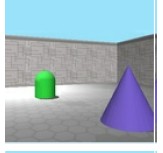

There is a green capsule behind a purple cone. The capsule is to the left of the cone.

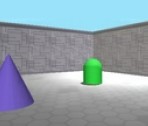

There is a purple cone to the left of a green capsule. The cone is in front of the capsule.

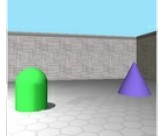

There is a green capsule in front of a purple cone. The capsule is to the left of the cone.

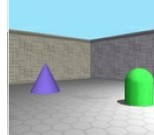

There is a green capsule to the right of a purple cone. The capsule is in front of the cone.

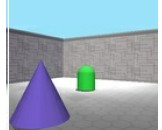

There is a purple cone in front of a green capsule. The cone is to the left of the capsule.

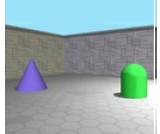

There is a green capsule to the right of a purple cone. The capsule is in front of the cone.

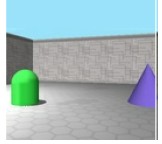

There is a green capsule in front of a purple cone. The capsule is to the left of the cone.

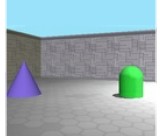

There is a green capsule in front of a purple cone. The capsule is to the right of the cone.

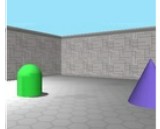

There is a green capsule to the left of a purple cone. The capsule is behind the cone.

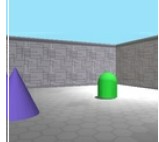

There is a green capsule behind a purple cone. The capsule is behind the cone.

### B.1.2 SYNTHETIC LANGUAGE, THREE OBJECTS

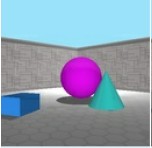

There is a large magenta sphere behind a aqua cone. The sphere is behind the cone. There is a aqua cone to the right of a blue box. There is a blue box in front of a large magenta sphere.

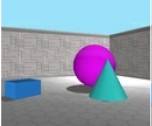

There is a aqua cone in front of a large magenta sphere. There is a aqua cone to the right of a blue box. There is a large magenta sphere to the right of a blue box.

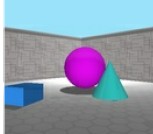

There is a large magenta sphere behind a aqua cone. The sphere is behind the cone. There is a aqua cone to the right of a blue box. There is a large magenta sphere behind a blue box. The sphere is behind the box.

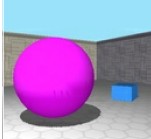

There is a aqua cone behind a large magenta sphere. The cone is behind the sphere. There is a aqua cone to the left of a blue box. There is a blue box to the right of a large magenta sphere.

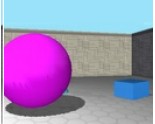

There is a large magenta sphere in front of a aqua cone. There is a aqua cone to the left of a blue box. There is a large magenta sphere to the left of a blue box.

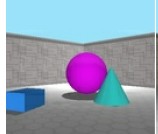

There is a aqua cone in front of a large magenta sphere. There is a aqua cone to the right of a blue box. There is a large magenta sphere behind a blue box. The sphere is behind the box.

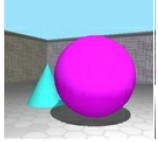

There is a aqua cone to the left of a large magenta sphere. There is a blue box to the right of a aqua cone. There is a blue box behind a large magenta sphere. The box is behind the sphere.

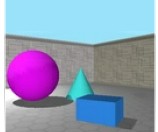

There is a aqua cone to the right of a large magenta sphere. There is a blue box in front of a aqua cone. There is a large magenta sphere to the left of a blue box.

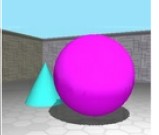

There is a aqua cone to the left of a large magenta sphere. There is a aqua cone to the left of a blue box. There is a blue box behind a large magenta sphere. The box is behind the sphere.

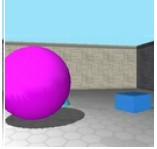

There is a aqua cone behind a large magenta sphere. The cone is behind the sphere. There is a blue box to the right of a aqua cone. There is a large magenta sphere to the left of a blue box.

### B.1.3 Natural language, two objects

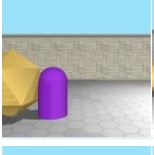

a room has two grey walls and a light blue ceiling. a large yellow cylinder is on the left next to a pink tube in the center.

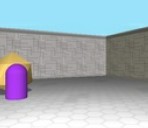

Corner of a room with a blue ceiling grey brick walls and grey tile floor. There are two objects in the room and the are on the left side of the room furthest from the corner. Object one is a medium size purple dome and it is in front of the larger medium sized gold double pointed round cylinder. The room is three dimensional as well as the objects.

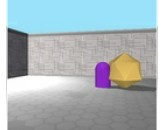

in grey room, under a blue sky a tan ball sits next to a purple rounded cylinder against a grey brick wall.

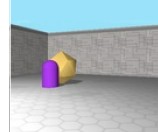

a room has two grey walls and a light blue ceiling. a yellow ball is in the center.

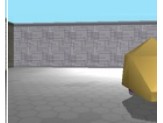

In this image a gold dodecagon is in the right corner.

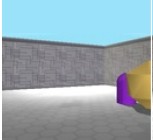

On the very right side of the room, in the center, is a tan prism shape with the bottom resembling a sphere, only the left half is visible. It is roughly half the size of the wall behind it. Behind that shape is a purple rod shape that is shorter. The right side is partially obscured by the shape in front of it.

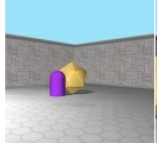

A purple shape that is like the end of a hotdog. Shape with 3 triangles on top and bottom to form points and 8 triangles that round out the middle

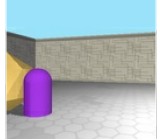

To the far left is a small purple bullet with a tan 10 sided shape to the left border.

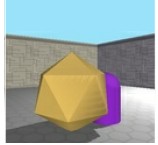

A yellow decagon sits in front of a purple cylinder which ends with a dome. Walls of grey brick converge behind them. Floor is grey octagons.

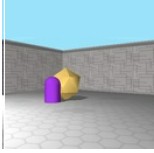

Corner of a room, blue ceiling with grey brick walls and a grey tiled floor, There are two objects in the room and towards the back corner. Object one is a purple slender dome and behind it slightly to the right is a gold hexagon. Both objects are 3 dimensional.

### B.1.4  NATURAL LANGUAGE, THREE OBJECTS

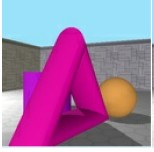

There is a huge lavender triangle on the left side of the room and a purple cube behind it on the left side and a light brown ball on the right side of it.

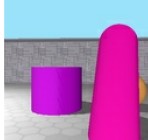

In the middle of the left side of the frame is a purple cylinder shape. Close up on the right side of the screen is a pink rounded cone almost sphere like at the top. It about an inch taller than the cylinder. Behind the pink shape is a minor protrusion of an orange sphere it comes out about halfway down the pink shape and barely pokes out (.25 in)

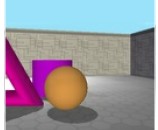

There is a yellow sphere in the middle of the image. Behind the sphere to the left, is a purple cube the majority of which is covered by sphere. There is a hot pink three dimensional triangle made of three dimensional cylinders, with an open center. Half of this triangle is covered by the left.

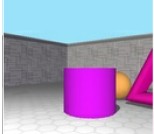

The ceiling is sky blue. The floor has cream-colored, six-sided tiles throughout. There are two side-half walls the same color as the floor. There is a neon-purple colored wide cylinder shape in the center of the floor, behind that and slightly to the right is a yellow shaped semicircle shape, and to the right of that is a neon pink shape of part of a triangle slightly showing.

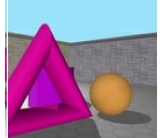

a light grey room has a light blue ceiling. a large pink triangle is in front left and yellow ball is in the center.

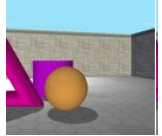

There is a pink triangle to the left of a yellow ball. behind them is a pink cylinder.

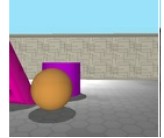

an oranger sphere sits in grey room next to a pink triangle and pink cylinder.

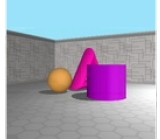

It's a grey brick room and floor with three geometric shapes in the center. There is a purple cylinder, pink triangle behind that, and a yellow ball in the rear.

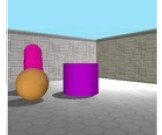

pink rounded at the top cylinder object in front of it is a orange ball to the right is a purple big round cylinder slightly taller then the ball. baby blue wall on top grey boarder horizontal then light grey under boarder. the floor is grey pattern octagon shaped.

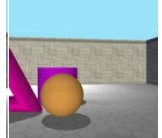

In the left front center of the screen is a tan ball with a flat dome shape peeking from behind and a giant hollow purple triangle half visible on the left of the screen.

## B.2 EXAMPLES OF SYNTHETIC CAPTION NOT MATCHING THE IMAGE

Examples where human annotators judge the synthetic caption to not match, or only partially match, the gold image. This shows imperfections of generated synthetic language.

| | Majority Human Label | Synthetic Description |
|---|---|---|
|  | No | There is a yellow torus behind a large yellow box. The torus is behind the box. *(The torus would be described to the right, in a more natural way. Also, the torus is on the border between yellow and green, and would more likely be described as green to contrast the other object.)* |
|  | No | There is a yellow torus to the left of a purple ico. There is a large lime capsule to the right of a yellow torus. There is a large lime capsule in front of a purple ico. *(The ico is occluded.)* |
|  | No | There is a peach torus in front of a large purple capsule. The torus is to the right of the capsule. *(The object at the back is occluded.)* |
|  | No | There is a purple cone far away. *(The cone is occluded by an object that is mostly out of the view.)* |
|  | Partial | There is a yellow cone far away. *(Object that is mostly out of the view is not mentioned.)* |
|  | Partial | There is a aqua torus to the right of a mint ico. *(The shape of the aqua object is ambiguous given its rotation.)* |

