# OpenReview forum: "Learning to encode spatial relations from natural language"
_ICLR.cc/2019/Conference_

### Official Review · AnonReviewer3 · 2018-11-01
**A new dataset for 3D understanding**

**Rating:** 5
**Confidence:** 4

**Review:**

The authors present a large synthetic dataset for 3D scenes with templated descriptions.  They then use the model of Eslami 2018 to this new domain.  The previous work appears to already introduce all the necessary mechanisms for 3D generalization from multiple viewpoints, though this work embeds language instead of a scene in the process.  Minor note: A bit more discussion on this distinction would be appreciated.  Also, it appears that the previous work includes many of the rendered scenes also present here, so the primary focus of this paper is on the use of a language encoder (not necessarily a trivial extension).

The model appears to perform well with synthetic data though very poorly with natural sentences.  This may be in part due to the very small dataset size.  It would be helpful to know how much of the performance gap is due to scaling issues (10M vs 5.6K) versus OOVs, new syntactic constructions, etc.  In particular, the results have ~two deltas of interest (NL vs SYN) and the gap in the upper bound from 0.66 to 0.91.  What do new models need to be able to handle to close these gaps?

Regarding the upper bound, there is some discussion that annotators might have had a strict definition of a perfect match.  Were annotators asked about this? The current examples (B.2), as the authors note, are more indicative of failings with the synthetic language than the human annotators.  This may again be motivation for collecting more natural language which would resolve some of the ambiguity and pragmatics of the synthetic dataset.

It would also be helpful to have some ablations included in this work. The most obvious being the role of $n$ (number of scene perspectives).  How crucial is it that the model has access to 9 of 10 perspectives?  One would hope that given the limited set of objects and colors, the model would perform well with far fewer examples per scene, learning to generalize across examples.

Since the primary contributions of the paper are a language dataset and a language encoder for the existing model of Eslami 2018, those should be discussed and ablated in the paper rather than relegated to the appendix.

Minor note:  the related work mentions grounding graphs which are core to work from Tellex and Roy, but omits existing fully neural end-to-end models in grounding (e.g. referring expressions work).

---

> ### Author Response · Authors · 2018-11-27
> **Response to reviewer3**
>
> Thank you for the thorough review and insightful comments. To address each comment individually:
>
> >> ‘A bit more discussion on this distinction would be appreciated.  Also, it appears that the previous work includes many of the rendered scenes also present here, so the primary focus of this paper is on the use of a language encoder (not necessarily a trivial extension).’ Thank you for this suggestion. We have modified the text to stress which aspects of this work are novel. When it comes to the model aspect of the paper, the use of a language encoder is the main modification. However, the primary focus of our paper is the analysis of the representations the model learns.
>
> >> ‘It would be helpful to know how much of the performance gap is due to scaling issues (10M vs 5.6K) versus OOVs, new syntactic constructions, etc.  In particular, the results have ~two deltas of interest (NL vs SYN) and the gap in the upper bound from 0.66 to 0.91.  What do new models need to be able to handle to close these gaps?’ Indeed the main obstacle with natural language is the small size of the dataset, which we can see on improvement from SLIM (NL) to SLIM (NL+SYN) in Figure 4, where we used additional synthetic data. Investigating the further improvements additional natural language data might have would require further data collection which is costly, however, collecting a larger NL dataset or generating richer synthetic language would likely contribute towards closing the gap. The gap in the upper bound (0.66 to 0.91) is an artifact of the task, and matching these numbers would mean the model has reached the human performance in interpreting the input descriptions.
>
> >> ‘Were annotators asked about this? The current examples (B.2), as the authors note, are more indicative of failings with the synthetic language than the human annotators. This may again be motivation for collecting more natural language which would resolve some of the ambiguity and pragmatics of the synthetic dataset.‘ For the synthetic dataset, the noise added by certain objects missing from the captions does not degrade the performance of the model - the model is *only* fed captions as input which means it can integrate the information of multiple captions to create a reconstruction of the scene, and it only reconstructs image pixels, which means it never sees that picture’s caption. Therefore the imperfect match between pixels and text is an issue that only arises at [human] evaluation time. Given the cost of cleaning up these captions, we believe the most practical course of action was to collect human evaluations with this mapping and keeping in mind that the gold truth sets an upper bound.
>
> >> ‘It would also be helpful to have some ablations included in this work. The most obvious being the role of $n$ (number of scene perspectives).  How crucial is it that the model has access to 9 of 10 perspectives?’ For non-language related hyperparameters, we followed the choices in Eslami et. al 2018. We choose to train by providing 9 input captions, but at test time we can control how many views of the scene the model gets or additionally sample them only from a restricted view of the room. In figure 7 you can find the results of that investigation where in the top row we show the output of the model when it has access to only a single view of the scene, and bottom row where it has access to multiple views of the scene, but from a restricted set of angles. We hope this analysis resolves your concerns but are happy to hear more specific suggestions as to further investigation.
>
> >> ‘Since the primary contributions of the paper are a language dataset and a language encoder for the existing model of Eslami 2018, those should be discussed and ablated in the paper rather than relegated to the appendix.’ Due to the page limitation we have provided a summary of the model changes and a description of the dataset, as the main focus of the current work is the representation analysis. However we understand we could have provided more detail in the main text. We have updated the text to describe the novel aspects of the model as well as dataset generation as much as possible. If there is any concrete information you feel is missing in the main text which would improve the text we would be very happy to hear!
>
> >> ‘Minor note:  the related work mentions grounding graphs which are core to work from Tellex and Roy, but omits existing fully neural end-to-end models in grounding (e.g. referring expressions work).’ Thank you for suggesting to include that line of work in our related works discussion. We have made appropriate amendments to note the referring expressions literature and how this relates to our work.

---

### Official Review · AnonReviewer1 · 2018-11-02
**Learning to encode spatial relations from natural language**

**Rating:** 5
**Confidence:** 4

**Review:**

The main contributions of the work are the new datasets and the overall integration of previous modeling tools in such a way that the final architecture is able to encode semantic spatial relations from textual descriptions. This is demonstrated by an implementation that, given textual descriptions, is able to render images from novel viewpoints. In terms of these two contributions, as I explain below, I believe there is space to improve the datasets and the paper needs further analysis/comments about the merits of the proposed approach. So my current overall rating is below acceptance level.

In terms of data, authors provide 2 new datasets: i) a large datasets (10M) with synthetic examples (images and descriptions) and ii) a small dataset (6k) with human textual descriptions corresponding to synthetic images. As the main evaluation method of the paper, the author include direct human evaluation of the resulting renderings (3 level qualitative evaluation: perfect-match/partial-match/no-match). I agree that, for this application, human evaluation is more adequate than comparing a pixel-level output with respect to a gold image. In this sense, it is surprising that for the synthetic dataset the perfect match score of human evaluation for ground truth data is only 66%. It will be good to increase this number providing a cleaning dataset.

Related to the previous comment, it will be good to provide a deeper analysis about the loss function used to train the model.

In terms of the input data, it is not clear how the authors decide about the 10 views for each scene.

In terms of the final model, if I understood correctly, the paper does not claim any contribution, they use a model presented in a previous work (actually information about the model is mostly included as a supplemental material). If there are relevant contributions in terms of model integration and/or training scheme, it will be good to stress this in the text.

Writing is correct, however, authors incorporate important details about the dataset generation process as well as the underlying model in the supplemental material. Given that there is a page limit, I believe the relevant parts of the paper should be self-contain.

---

> ### Author Response · Authors · 2018-11-27
> **Response to reviewer1**
>
> Thank you for the thorough review and insightful comments.
>
> >> ‘In this sense, it is surprising that for the synthetic dataset the perfect match score of human evaluation for ground truth data is only 66%. It will be good to increase this number providing a cleaning dataset.’ For the synthetic dataset, the noise added by certain objects missing from the captions does not degrade the performance of the model - the model is *only* fed captions as input which means it can integrate the information of multiple captions to create a reconstruction of the scene, and it only reconstructs image pixels, which means it never sees that picture’s caption. Therefore the imperfect match between pixels and text is an issue that only arises at [human] evaluation time. Given the cost of cleaning up these captions, we believe the most practical course of action was to collect human evaluations with this mapping and keeping in mind that the gold truth sets an upper bound.
>
> >> ‘deeper analysis about the loss function used to train the model.’ We use the loss function from previous work by Eslami et al. (2018)--. We have added more details to the model section in the main text and also provide a detailed description in Appendix section A.2.
>
> >> ‘In terms of the input data, it is not clear how the authors decide about the 10 views for each scene.’ For non-language related hyperparameters, we followed the choices in Eslami et. al 2018. Optimizing this hyperparameter choice is an interesting avenue to explore however we felt it was outside of the scope of this work.
>
> >> ‘If there are relevant contributions in terms of model integration and/or training scheme, it will be good to stress this in the text.’  We have updated the text to explain the novel aspects of the model, which is the integration of language inputs, as well as dataset generation as much as possible. In the appendix you can also find a detailed description of the model setup, hyperparameters, and dataset generation which did not fit into the main text.

---

### Official Review · AnonReviewer2 · 2018-11-03
**Good paper but needs better positioning and presentation**

**Rating:** 6
**Confidence:** 5

**Review:**

This paper presents a system to map natural language descriptions of scenes containing spatial relations to 3D visualizations of the corresponsing scene. The authors collect a dataset of different scenes containing objects of varying shapes and colors, along with several descriptions from different viewpoints. They train a model based on the Generative Query Network, to generate scenes conditioned on multiple text descriptions as input, along with associated camera angles. Empirical results using human evaluators demonstrate better performance compared to baselines and the authors perform a good analysis of the model, showing that it learns to ground the meaning of spatial words robustly.

Pros:
1. Well-executed paper, with convincing empirical results on the newly collected dataset.
2. Nice analysis to demonstrate that the model indeed learns good semantic representations for spatial language.

Cons:
1. The positioning of this paper with respect to recent work is disappointing. In both the Introduction and Related Work sections, the authors talk about dated models for spatial reasoning in language (pre-2012). There have been several pieces of work that have looked at learning multimodal representations for spatial reasoning. These are a few examples:
   a) Misra, Dipendra, John Langford, and Yoav Artzi. "Mapping Instructions and Visual Observations to Actions with Reinforcement Learning." Proceedings of the 2017 Conference on Empirical Methods in Natural Language Processing. 2017.
   b) Michael Janner, Karthik Narasimhan, and Regina Barzilay. "Representation Learning for Grounded Spatial Reasoning." Transactions of the Association of Computational Linguistics 6 (2018): 49-61.
   c) Paul, Rohan, et al. "Grounding abstract spatial concepts for language interaction with robots." Proceedings of the 26th International Joint Conference on Artificial Intelligence. AAAI Press, 2017.
   d) Ankit Goyal, Jian Wang, Jia Deng. Think Visually: Question Answering through Virtual Imagery. Annual Meeting of the Association for Computational Linguistics (ACL), 2018

Even though Gershman & Tenenbaum (2015) demonstrate weaknesses of a specific model, some of the above papers demonstrate models that can understand things like "A is in front of B" = "B is behind A". A discussion of how this paper relates to some of this prior work, and an empirical comparison (if possible) would be good to have.

2. The introduction reads a bit vague. It would help to clearly state the task considered in this paper upfront i.e. generating 3D scenes from text descriptions at various viewpoints. In the current form, it is hard to understand the task till one arrives at Section 3.

Other comments:
1. What is the difference between the bar graphs of Figures 5 and 6? Is the one on Figure 5 generated using the sentences (and their transforms) from Gershman & Tenenbaum? If so, how do you handle unseen words that are not present in your training data?
2. Would be helpful to clearly explain what the red and black arrows represent in Figure 7.
3. How does the model handle noisy input text i.e. if the object descriptions (shape/color) are off or if some of the input text is incorrect (say a small fraction of the different viewpoints)?

------
Edit:
Thank you for the author response. Even if you consider the story to be the same across literature (which in this case is not, since the more recent models handle spatial relations that the previous ones failed on), it's still worth doing due diligence to the recent work, especially so that the reader gets a better sense of how to position your work amongst these.

---

> ### Author Response · Authors · 2018-11-27
> **Response to reviewer2**
>
> Thank you for the thorough review and insightful comments. To address each comment individually:
> 1. Thank you for suggesting additional literature to consider. We have updated the paper to incorporate more of this more recent work and have clarified the connection to our work. While we acknowledge that some of those papers are worth discussing in the context of our work, the story remains unchanged, as these more recent papers ended up using the same simplifications or short-cuts when considering spatial relations that we already discussed in relation to earlier publications: spatial relations are either treated as absolute, well-defined metrics in discrete (block world) spaces, or in the continuous case of the robotics work, the complexity of real world data and relations is simplified through manually defined mappings onto symbolic, abstract factor graphs, which again then simplifies the aspect of spatial relations that we are focused on learning here.
> 2. We have added a section to the introduction to better introduce the task.
>
> Comments:
> 1. Figure 5 compares the similarities for a single sentence as encoded by the encoder. Figure 6 on the other hand compares the similarity for the scene-level representation, after the aggregation step. We were interested in studying the representations at the two stages, as the representation studied in the first is more directly comparable to classic NLP models, while the second integrates information from multiple perspectives. This allows us to show that the spatial invariances in language are detected already at the encoder level, and are not due to the aggregation.
> 2. We have clarified this in the caption: black is the camera angle fed to the encoder to reconstruct, while red is the target camera angle.
> 3. This is a great question! In fact this was the question we sought to address by collecting the natural language dataset, and we have observed that many human annotations are unusually written or employ rare words. We can see that in spite of variation in specific descriptions, the integration over different captions confers extra robustness to the model and can still generate consistent scenes for this natural language setup, as evidenced by figure 3.

---

### Meta-Review · Area_Chair1 · 2018-12-13
**Promising work, but just below threshold in its current form**

**Confidence:** 4
**Recommendation:** Reject

**Metareview:**

Strengths: Execution of paper well received. Results on new dataset. Convincing demonstration that the proposed approach learns good semantic representations.

Weaknesses: Reviewers felt the positioning with prior work was not as strong as it could be. Reviewers wanted to have seen an ablation study.

Contention: Some general agreement among both the one positive reviewer and negative reviewer that the representation of prior work is skewed.

Consensus: With two 5s and one 6 the numerical average of 5.33 is representative of the aggregated consensus opinion which is that the work is just below threshold in its current form.